

# Relative humidity-dependent viscosity of secondary organic material from toluene photo-oxidation and possible implications for organic particulate matter over megacities

M. Song[1,2], P. F. Liu[3], S. J. Hanna[1], R. A. Zaveri[4], K. Potter[5], Y. You[1], S. T. Martin[3,6], A. K. Bertram[1]

[1] {Department of Chemistry, University of British Columbia, Vancouver, BC, V6T 1Z1, Canada}

[2] {Department of Earth and Environmental Sciences, Chonbuk National University, Jeollabuk-do, Republic of Korea}

[3]{John A. Paulson School of Engineering and Applied Sciences, Harvard University, Cambridge, Massachusetts 02138, USA}

[4]{Atmospheric Sciences and Global Change Division, Pacific Northwest National Laboratory, Richland, WA, USA}

[5]{School of Chemistry, University of Bristol, Bristol, UK BS8 1TS, UK}

[6]{Department of Earth and Planetary Sciences, Harvard University, Cambridge, Massachusetts 02138, USA}

Correspondence to: A. K. Bertram (bertram@chem.ubc.ca)

## Abstract

To improve predictions of air quality, visibility, and climate change, knowledge of the viscosities and diffusion rates within organic particulate matter consisting of secondary organic material (SOM) is required. Most qualitative and quantitative measurements of viscosity and diffusion rates within organic particulate matter have focused on SOM particles generated from biogenic VOCs such as α-pinene and isoprene. In this study, we quantify the relative humidity (RH)-dependent viscosities at $295 \pm 1$ K of SOM produced by photo-oxidation of toluene, an anthropogenic VOC.





The viscosities of toluene-derived SOM were $2 \times 10^{-1}$ to ~ $6 \times 10^{6}$ Pa·s from 30 to 90% RH, and
greater than ~$2 \times 10^{8}$ Pa·s (similar to or greater than the viscosity of tar pitch) for RH ≤ 17%. These
viscosities correspond to Stokes-Einstein-equivalent diffusion coefficients for large organic
molecules of ~$2 \times 10^{-15}$ cm²·s⁻¹ for 30% RH, and lower than ~$3 \times 10^{-17}$ cm²·s⁻¹ for RH ≤ 17%.
Based on these estimated diffusion coefficients, the mixing time of large organic molecules within
200 nm toluene-derived SOM particles is 0.1 - 5 hr for 30% RH, and higher than ~100 hr for RH
≤ 17%. These results were used, as a first-order approximation, to estimate if organic particulate
matter will be in well-mixed over the world's top 15 most populous megacities. If the organic
particulate matter in the megacities is similar to the toluene-derived SOM in this study, in Kolkata,
Istanbul, Dhaka, Tokyo, Shanghai, and Mumbai, mixing times in organic particulate matter during
extended periods of the year will be very short, and well-mixed particles can be assumed. On the
other hand, the mixing times of large organic molecules in organic particulate matter in Delhi,
Beijing, Mexico City, Cairo, and Karachi may be long and the particles may not be well-mixed in
the afternoon (3:00 – 5:00 local time) during certain times of the year.
**1 Introduction**
Volatile organic compounds (VOCs) are released into the atmosphere from both biogenic and
anthropogenic sources. In the atmosphere, VOCs can form secondary organic material (SOM)
through oxidation reactions with OH radicals, $NO_3$ radicals, and $O_3$. SOM accounts for 20 – 80%
of the mass of organic atmospheric particulate matter at various locations (Zhang et al., 2007;
Jimenez et al., 2009). SOM typically consist of thousands of different compounds, and only 10 –
20% of the individual molecules that make up SOM particles have been identified (Decesari et al.,
2006; Hallquist et al., 2009). The lack of information on the chemical composition of SOM has
resulted in a poor understanding of their physical properties, including the viscosity and molecular
diffusion rates within SOM particles.
Knowledge of the viscosity and molecular diffusion rates within SOM particles is needed to predict
the properties of these particles and understand their role in the atmosphere. For example, the size
distribution and mode diameter depend on the diffusion rates of organic molecules within the
particles (Riipinen et al., 2011; Zaveri et al., 2014). Simulations show that total SOM mass





concentrations can be overestimated or underestimated depending on what diffusion rates are used
(Shiraiwa and Seinfeld, 2012). Chemical aging of atmospheric particles by heterogeneous
reactions can depend on diffusion rates within SOM (Shiraiwa et al., 2011; Kuwata and Martin,
2012; Zhou et al., 2013; Steimer et al., 2014; Houle et al., 2015) and heterogeneous ice nucleation
may be influenced by the viscosity of SOM particles (Murray et al., 2010; Wang et al., 2012;
Ladino et al., 2014; Schill et al., 2014; Wilson et al., 2012). Moreover, long-range transport of
polycyclic aromatic hydrocarbons can depend on diffusion rates in a particle (Zelenyuk et al., 2012;
Zhou et al., 2013) and the efflorescence of crystalline salts can be hindered for highly viscous
SOM (Murray, 2008; Murray and Bertram, 2008; Bodsworth et al., 2010; Song et al., 2013).
Most qualitative and quantitative measurements of viscosity and diffusion rates within organic
particulate matter have focused on SOM generated from biogenic VOCs such as α-pinene and
isoprene (Virtanen et al., 2010; Cappa et al., 2011; Perraud et al., 2012; Saukko et al., 2012;
Abramson et al., 2013; Robinson et al., 2013; Renbaum-Wolff et al., 2013a; Bateman et al., 2015;
Kidd et al., 2014; Pajunoja et al., 2014; Wang et al., 2014; Song et al., 2015). Recently, the
viscosity and diffusion rates within SOM particles generated from anthropogenic VOCs have also
been investigated. Using single particle mass spectrometry, Robinson et al. (2013) investigated
mixing of toluene-derived SOM particles and SOM particles from α-pinene ozonolysis. One
possible explanation of the lack of mixing within toluene-derived SOM particles was a high
viscosity of the SOM. From bounce experiments, Saukko et al. (2012) reported that SOM particles
derived from naphthalene and n-heptadecane are highly viscous upon increasing oxidation. Also
from bounce experiments, Bateman et al. (2015) showed SOM derived from photo-oxidation of
toluene had a viscosity > 100 Pa·s for relative humidity (RH) values < 80%. Li et al. (2015)
showed through bounce experiments that SOM derived from $m$-xylene and 1,3,5-trimethylbenzene
had a viscosity of > 100 Pa·s at RH values less than 70%. Li et al. (2015) also used results of
reactive uptake studies to infer that for RH values of 35 - 45% the diffusion coefficient of
carboxylic acids within SOM generated from several anthropogenic VOCs (toluene, m-xylene and
1,3,5-trimethylbenzene) was $3 \times 10^{-18 \pm 0.5}$ m$^2$ s$^{-1}$. Although there has been recent progress in
measuring the viscosity and diffusion rates within SOM generated from anthropogenic VOCs,
additional studies are needed to quantify the viscosities and diffusion rates over the full range of
RH found in the atmosphere.



In the following, we measure the viscosities of toluene-derived SOM over the range of RH values
found in the atmosphere. As in previous studies, SOM from the photo-oxidation of toluene serves
as a proxy for organic particulate matter from anthropogenic sources in megacities (e.g. Pandis et
al., 1992; Robinson et al., 2013). After determining viscosities as a function of RH, the Stokes-
Einstein equation is used to convert the viscosities into equivalent diffusion rates of large organic
molecules within toluene-derive SOM. The Stokes-Einstein equation should give reasonable
values of diffusion rates when the viscosity is not near the viscosity of a glass (~$10^{12}$ Pa s) and
when the molecules are roughly the same size or larger than the molecules in the SOM matrix
(Champion et al., 2000; Koop et al., 2011; Shiraiwa et al., 2011; Power et al., 2013). Finally, the
results are used to estimate the viscosities and diffusion rates in organic particulate matter over
megacities.

## 2 Experimental

The production and collection of SOM particles onto hydrophobic substrates (which are needed
for the beam mobility and poke-and-flow experiments) is discussed in Sect. 2.1. The viscosity of
toluene-derived SOM was determined using the bead-mobility technique and the poke-flow
technique together with simulations of fluid flow. These two techniques are discussed in Sects. 2.2
and 2.3.

## 2.1 Production and collection of secondary organic material on hydrophobic substrates

SOM aerosols particles having diameters less than 1 μm were generated by toluene photo-
oxidation in an oxidation flow reactor (OFR) (Kang et al., 2007; Lambe et al., 2011). The
procedure for generating SOM from toluene photo-oxidation in the flow reactor has been given by
Liu et al. (2013). Only the details relevant to the current experiments are given here.
For this study, the volume of the OFR was 13.3 L and the reactor was operated at a flow rate of
~7 L m$^{-1}$ with a residence time in the range of ~ 110 s. The temperature used in the OFR
experiments was 293 ± 2 K and the concentrations of toluene and ozone used in the flow reactor
are listed in Table 1. Ozone was produced external to the flow reactor by irradiating pure air with





the ultraviolet emission from an Hg lamp (λ = 185 nm). The injected ozone concentration was ~30
ppm. Hydroxyl radicals were produced inside the OFR by the following photochemical reactions:
$O_3 + h\nu$ (λ = 254 nm) $\longrightarrow$ $O_2 + O(^1D)$                (R1),
$O(^1D) + H_2O \longrightarrow 2OH$                    (R2)
Mass concentrations of SOM particles in the OFR were 60-100 µg m$^{-3}$ and 600-1000 µg m$^{-3}$ for
the two different experimental conditions (see Table 1). At the outlet of the OFR, two different
methods were used for the collection of SOM particles. In the first method, SOM particles were
collected on a hydrophobic slide using an electrostatic precipitator (TSI 3089, USA). After
collection, the SOM particles on the hydrophobic slides, formed from coalescence during sampling,
were smaller than ~5 µm in diameter. For the bead-mobility and poke-flow techniques, however,
particle sizes between 20 - 60 µm in diameter are needed.  To generate these large sizes, the
hydrophobic slides containing the SOM particles were placed inside an RH- and temperature-
controlled flow cell (Pant et al., 2006; Bertram et al., 2011; Song et al., 2012) and the RH was
increased to > 100%. This procedure caused particle growth by water uptake and eventual
coagulation among particles. This growth and coagulation process resulted in larger but fewer
SOM particles on the hydrophobic slides. Details of this procedure are given by Renbaum-Wolff
et al. (2015) and Song et al. (2015). This procedure was used for samples 1, 2, 5, and 6 shown in
Table 1.
In the second method, SOM particles were collected on hydrophobic surfaces using a single stage
impactor (Prenni et al., 2009; Poschl et al., 2010). During impaction, the collected submicron SOM
particles coagulated, resulting in particles with sizes between 10 and 100 µm in diameter. These
supermicron particles were used directly in the bead-mobility and poke-flow experiments. This
procedure was used to collect samples 3, 4, 7, and 8 shown in Table 1.
For all the bead-mobility experiments, a Teflon substrate was used. For all the poke-and-flow
experiments, hydrophobic glass slides coated with trichloro(*1H,1H,2H,2H*-perfluorooctyl)silane
(Sigma-Aldrich) were used. The coating procedure is described in Knopf (2003).
**2.2 Bead-mobility experiments**





The bead-mobility technique was previously described in Renbaum-Wolff et al. (2013a, b). Briefly,
a water suspension of ~1 μm insoluble melamine beads (Sigma Aldrich Cat. #86296) was
nebulized and incorporated into supermicron SOM particles deposited on a hydrophobic substrate
(toluene samples 1-4, Table 1). The hydrophobic substrate with the SOM particles and beads was
placed in a flow-cell with variable RH and a temperature of 295 ± 1 K. A continuous flow of
$N_2/H_2O$ gas (flow rate ≈ 1200 sccm) was passed over the supermicron particles. The flow above
the particles resulted in a shear stress on the particle surface and internal circulations within the
particle, which could be visualized by observing the beads within the particles with a light-
transmitting microscope coupled to a CCD camera (Zeiss Axio Observer, magnification 40×).
Figure 1 shows images from a typical bead-mobility experiment for a toluene-derived SOM
particle at 80% RH. Typically, 1 to 7 beads were monitored within a particle over 50 - 100 frames.
The time between frames ranged from 0.2 s – 10 min depending on the velocity of the beads. From
the location of the beads as a function of time, the speed of individual beads was determined. These
individual speeds were then used to determine average bead speeds for a given sample and RH.
The measured speeds of 3 - 10 beads were used to determine a mean bead speed. Bead speeds were
not reported at RH < 60% since at these RH values the movements of the beads were too slow to
measure for typical observation times.
The average bead speed for a given sample and RH was converted to viscosity using a calibration
line. The calibration line was developed by Song et al. (2015) from measurements of bead speed
in sucrose-water particles over a range of RH values. The RH within the flow-cell was measured
using a hygrometer with a chilled mirror sensor (General Eastern, Canada), which was calibrated
by measuring the deliquescence RH for pure ammonium sulfate particles (80.0% RH at 293 K,
Martin (2000)). The uncertainty of the hygrometer was ± 0.5% RH after calibration.
**2.3 Poke-flow technique in conjunction with fluid simulation**
The poke-and-flow technique in conjunction with fluid simulations was used to measure the
viscosities of SOM particles at RH values less than 50%. This technique was not used at RH values
> 50% since the flow rates of the SOM after poking were too fast to observe at these RH values.
The qualitative method of poking an inorganic particle to determine its phase (i.e., solid or liquid)
was introduced by Murray et al. (2012). Renbaum-Wolff et al. (2013a) and Grayson et al. (2015a)
expanded on this method by measuring the characteristic flow time of a material after poking and





extracting viscosity information from simulations of fluid flow. Briefly, supermicron toluene-
derived SOM particles deposited on a hydrophobic substrate (Toluene 5 to 8, Table 1) were placed
inside a flow-cell with RH control. The particles were conditioned for 30 min at > 70% RH, 60
min at $60 - 70\%$ RH, 2 h at $30 - 60\%$ RH, and 3 h at $\leq 30\%$ RH. These times are sufficient for the
particles to equilibrate with the surrounding water vapor based on recent measurements of
diffusion coefficients of water within SOM (Price et al., 2015). After equilibration, particles were
poked using a sharp needle (0.9 mm × 40 mm) (Becton-Dickson, USA) that was mounted to a
micromanipulator (Narishige, model MO-202U, Japan) and inserted through a small hole in the
top of the flow-cell. The geometrical changes before, during, and after poking a particle were
recorded by a reflectance optical microscope (Zeiss Axio Observer, 40× objective) equipped with
a CCD camera. At 30 - 50% RH the action of poking the particles with the needle resulted in the
material forming a half torus geometry (see Fig. 2a). From the images recorded after poking the
particles, the experimental flow time, $\tau_{exp,\ flow}$, was determined. The experimental flow time was
defined as the time taken for the equivalent-area diameter of the inside of the half torus geometry
to reduce to 50% of the initial diameter. Here the equivalent-area diameter, $d$, is calculated as $d =$
$(4A/\pi)^{1/2}$ where $A$ represents the hole area (Reist, 1992). For RH < 20% the SOM particles shattered
after poking, and no restorative flow was observed over ~5 hr (See Fig. 2b). In this case $\tau_{exp,\ flow}$
was set to > 5 hr.
To determine viscosities from $\tau_{exp,\ flow}$, simulations of fluid flow were carried out with the finite-
element analysis software package, *COMSOL Multiphysics* (Renbaum-Wolff et al., 2013a;
Grayson et al., 2015a).  The mesh size used in the simulations was $4.04 - 90.9$ nm. The physical
parameters (i.e., slip length, surface tension, contact angle, and density) used in the simulation are
listed in Table 2.
For each particle for which flow was observed, simulations were run using a half torus geometry,
similar to the geometry observed in the experiments where flow was observed. The radius of the
tube, $R$, in the half torus geometry and the radius of the hole, $r$, in the half torus geometry used in
the simulations were based on the images recorded immediately after poking the particles with the
needle. To determine viscosity for each particle, viscosity in the simulations was adjusted until
$\tau_{model,\ flow}$ was within 1% of $\tau_{exp,\ flow}$.
In cases for which the particles cracked, simulations were run using a quarter-sphere model with
one of the flat faces of the quarter sphere in contact with the substrate, similar to what was observed





in the experiments (Renbaum-Wolff et al., 2013a). The diameter used for the quarter sphere was
20 µm. In this case we determined a lower limit to the viscosity by adjusting the viscosity in the
simulation until the sharp edge of the quarter sphere model moved by 0.5 µm within 5 hr. A value
of 0.5 µm was used since this amount of movement could be observed in the optical microscope
experiments.

## 7   3 Results

Shown in Fig. 3 are the mean bead speeds of individual SOM samples (toluene 1, 2, 3, and 4)
measured at different RH values between 60 and 90% RH (see Sect. 2.1). As the RH decreased
from 89.9 to 60.7%, the average bead speed decreased by a factor of 22 from $9.20 \times 10^{-4}$ to $4.24 \times$
$10^{-7}$ µm·ms$^{-1}$. Sample-to-sample variation was less than the uncertainty in the measurements and,
within uncertainty, the results for 60 - 100 µg·m$^{-3}$ concentration agreed with the results for 600 -
1000 µg·m$^{-3}$ concentration.
Figure 4 shows the result of $\tau_{exp, flow}$ as a function of RH for the different samples (toluene 5, 6, 7,
and 8). The $\tau_{exp, flow}$ increased from ~1 s to ~2000 s as RH decreased from 50 to 30% RH. The $\tau_{exp,}$
$_{flow}$ variation from sample to sample was less than the variation within individual toluene samples
with mass concentrations over the range studied (600-1000 and 60-100 µg·m$^{-3}$), as shown in Fig.

18    4.

Shown in Fig. 5 are the viscosities as a function of RH for toluene-derived SOM particles
determined from the bead-mobility experiments (Sect. 2.2) and the poke-flow experiments in
conjunction with the fluid simulation (Sect. 2.3). For the bead-mobility experiments, the viscosities
were determined by the mean of bead speeds between 60 and 90% RH. The *y*-error bars indicate
the 95% prediction intervals from the calibration line (Song et al., 2015). The *x*-error bars represent
the uncertainty in the RH measurements. The viscosity of the SOM increases from ~0.2 to ~129
Pa·s as RH decreases from 89.9 to 60.7%. As shown in Fig. 5, difference between the results for
the 600-1000 and 60-100 µg·m$^{-3}$ samples are less than the uncertainties in the measurements.
Also shown in Fig. 5 are the calculated viscosities of the toluene-derived SOM for RH < 50% from
the poke-flow experiments. The viscosity increases from approximately $7.8 \times 10^{3}$ to $6.3 \times 10^{6}$ Pa·s





as RH decreases from 47.3 to 30.5%. The uncertainty in the viscosity of approximately two orders
of magnitude arises from the uncertainties in the physical parameters used in the simulations. For
RH < 20%, restorative flow did not occur over ~5 hrs resulting in a lower limit to the viscosity of
~$2 \times 10^8$ Pa·s, similar to or greater than the viscosity of tar pitch (~$10^8$ Pa·s, Koop et al. (2011)).

## 4 Discussion

Bateman et al. (2015) previously estimated the viscosity of toluene-derived SOM from particle
rebound experiments. From their measurements they estimated a viscosity of 100 to 1 Pa·s for
RHs between 60 and 80% with SOM mass concentrations of 30 - 50 µg·m$^{-3}$ (green box in Fig. 5)
in good agreement with our measurements.
Li et al. (2015) previously estimated the diffusion coefficient of carboxylic acids within toluene-
derived SOM from measurements of reactive uptake of NH$_3$. They estimated a diffusion
coefficient $10^{-17.5\pm0.5}$ m$^2$·s$^{-1}$ for RHs between 35 and 45% using SOM mass concentrations of 44 to
125 µg m$^{-3}$. If a hydrodynamic radius of 0.1 - 1.5 nm is assumed (Li et al., 2015), viscosity of $1 \times$
$10^4 - 2 \times 10^6$ Pa·s is calculated using the Stokes-Einstein equation (blue box in Fig. 5), consistent
with our measurements. The good agreement between the current results and the results from
Bateman et al. and Li et al. suggests that the viscosity of the toluene-derived SOM is relatively
insensitive to the particle mass concentrations at which the SOM is produced over the range of 30
to 1000 µg·m$^{-3}$.
A liquid is defined as a material with a viscosity less than $10^2$ Pa·s; a semisolid is defined as a
material with a viscosity between $10^2$ Pa·s and $10^{12}$ Pa·s; and a solid is defined as a material with
a viscosity greater than $10^{12}$ Pa·s (Koop et al., 2011; Shiraiwa et al., 2011). As shown in Fig. 5,
the viscosities of the SOM produced from toluene photo-oxidation correspond to liquid for RH >
60%, a semisolid for 60% < RH < 30%, and a semisolid or a solid for RH < 20%. Our results
suggest a semisolid-to-liquid phase transition at an RH between 60 and 70%, in good agreement
with Bateman et al. (2015) who suggested a semisolid-to-liquid phase transition of toluene-derived
SOM particles in the range of 60 - 80% RH.



From the viscosities determined at 295 ± 1 K and the Stokes-Einstein relationship (assuming a
hydrodynamic radius of 0.4 nm for organic molecules within the toluene-derived SOM, Renbaum-
Wolff et al., 2013a), we calculated the diffusion coefficients of large organic molecules, $D_{org}$,
within toluene-derived SOM (see secondary $y$-axis in Fig. 5). $D_{org}$ ranges from ~3 × 10$^{-8}$ to ~2 ×
10$^{-15}$ cm$^2$·s$^{-1}$ for RH from 90 to 30%. It is lower than ~3 × 10$^{-17}$ cm$^2$·s$^{-1}$ for RH ≤ 17%. The Stokes-
Einstein relation is not expected to predict with high accuracy the diffusion rates of small gas
molecules such as OH, $O_3$, $NO_x$, $NH_3$, and $H_2O$ (Koop et al., 2011; Shiraiwa et al., 2011). However,
the Stokes-Einstein relationship should give reasonable estimations of diffusion rates for large
organic molecules for conditions not close to the glass transition temperature of the matrix
(Champion et al., 2000; Koop et al., 2011; Shiraiwa et al., 2011; Power et al., 2013). However, the
relationship may be inaccurate near the glass transition RH (Champion et al., 2000; Shiraiwa et al.,
2011; Power et al., 2013).
Using the diffusion coefficients ($D_{org}$), the mixing time by diffusion, $\tau_{mixing}$, of large organic
molecules within a 200 nm SOM particle was calculated with the following equation, where $d$ is
the particle diameter (Shiraiwa et al., 2011; Bones et al., 2012; Renbaum-Wolff et al., 2013a):
$$\tau_{mixing} = \frac{d^2}{4\pi^2 D_{org}}$$   (Eq.1)
Here we are using 200 nm to represent a typical accumulation mode atmospheric particle, Shiraiwa
et al. (2011). The concentration of the diffusing molecules anywhere in the particles deviates by
less than $e^{-1}$ from the homogeneously well-mixed case at times longer than $\tau_{mixing}$. The $\tau_{mixing}$ values
calculated with this procedure are indicated in Fig. 5 (secondary $y$-axis). At an RH of 45% or
higher, the mixing times are short, approaching less than or equal to 0.1 h. At 30% RH the mixing
times are between 0.1 and 5 h. At RH ≤ 17% the mixing time is longer than ~100 h.
**5 Atmospheric implications**
In the following, we use the mixing times calculated in the previous section to estimate the mixing
times of large organic molecules in organic particulate matter over megacities. Several caveats
should be kept in mind when applying the mixing times discussed earlier to particles over
megacities. First, we are assuming that toluene-derived SOM is a good proxy for organic



particulate matter over megacities. Organic particulate matter over megacities are most likely more
complicated and may include inorganic components. Second, the toluene-derived SOM was
generated using relatively large mass concentrations of particles ($60 - 1000$ μg·m⁻³). The good
agreement between our results and the results from Bateman et al. (2015) and Li et al. (2015),
which were carried out with a mass concentration of $30 - 1000$ μg·m⁻³, suggests that for toluene-
derived SOM the viscosity is not strongly dependent on the mass concentration of organics used
to generated the SOM, but additional studies are needed to confirm this. Third, as mentioned above,
the Stokes-Einstein equation was used to estimate diffusion coefficients and hence mixing times,
and this equation can underestimate diffusion coefficients close to the glass transition temperature.
Despite these caveats, our estimates below should be a useful first-order approximation to the
mixing times of large organics in organic particulate matter over megacities.
For this analysis we define megacity as a metropolitan area with a total population in excess of ten
million people. Based on the Population Division Data Query (2014) of the United Nations
(http://esa.un.org/), we selected the top 15 most populous cities (Tokyo, Delhi, Shanghai, Mexico
City, São Paulo, Mumbai, Osaka, Beijing, New York, Cairo, Dhaka, Karachi, Buenos Aires,
Kolkata, and Istanbul) which meet this criterion.
In order to determine $\tau_{mixing}$ for organic particulate matter in megacities, information on the RH in
the cities is needed. Fig. 6 gives information on RH and temperature in the 15 most populous
megacities obtained from NOAA's National Climatic Data Center (NCDC) (www.ncdc.noa.gov).
The figure shows boxplots of average afternoon ($3:00 - 5:00$ local time) RH and temperature from
these cities for the years from $2004 - 2014$. The afternoon ($3:00 - 5:00$ local time) was chosen for
this analysis since this is the time of day when RH is typically the lowest. In the figure, the boxes
represent the median, 25[th] and 75[th] percentiles and the whiskers show the 10[th] and 90[th] percentiles.
In the Fig. 6, we indicate with green shading cases when the afternoon RH (at the 10[th] percentile
level) does not go below 45% RH. The cases when the afternoon RH (at the 10[th] percentile level)
does not go below 45 % RH are listed in Table 3 (second column).  At 45% RH the mixing time
within toluene-derived SOM is short (i.e., less than or equal to 0.1 h).  Figure 6 (green shading)
and Table 3 suggests that, if the organic particulate matter over megacities is similar to the toluene-
derived SOM in this study, in Kolkata, Istanbul, Dhaka, Tokyo, Shanghai, and Mumbai, mixing





times during extended periods of the year will be very short, and homogeneously well-mixed
particles can be assumed.
In the Fig. 6, we indicate with red shading cases where the afternoon RH (at the 10th percentile
level) is 17% or lower. The cases when the afternoon RH (at the 10th percentile level) is 17 % or
lower are listed in Table 3 (third column). As mentioned above, at this RH, the mixing time within
toluene-derived SOM is long (> 100 h), based on the viscosity measurements and Stokes-Einstein
calculations. Fig. 6 (red shading) and Table 3 (third column) suggests that if the organic particulate
matter is similar to the toluene-derived SOM in this study, in Delhi, Beijing, Mexico City, Cairo,
and Karachi, the particles may not be well-mixed in the afternoon (3:00 – 5:00 local time) during
certain times of the year. If the breakdown to the Stokes-Einstein relationship is large at high
viscosities the number of cases classified as having particles not well-mixed may be less than
indicated.  Hence, the number of cases classified as having particles not well-mixed based on the
viscosity data presented here and the Stokes-Einstein relationship should be considered as an upper
limit.
Another caveat to the discussion above is the effect of temperature on viscosity. The conclusions
reached above were based on viscosity measurements carried out at 295 ± 1 K, while the
temperatures shown in Fig. 6 ranges from roughly 0 °C to 40 °C.  Viscosity is known to decrease
as the temperature increases, but the effect of temperature on the viscosity of SOM has not been
quantified. Studies that quantify the effect of temperature on the viscosity of SOM are needed for
more accurate predictions of the mixing times in organic particulate matter over megacities.
Kleinman et al., (2009) studied the time evolution of aerosol size distributions and number
concentrations of ambient particulate matter over the Mexico City plateau during the MILAGRO
(Megacity Initiative: Local And Global Research Observations) field campaign conducted in
March 2006. The particulate matter over Mexico City was primarily organic and as photochemical
aging occurred, Kleinman and colleagues observed an increase in accumulation-mode volume due
to an increase in the accumulation mode particles, not because of an increase in the average size
of the accumulation mode. The condensing organic vapors from photooxidation of toluene and
other anthropogenic VOCs over Mexico City are expected to be semivolatile (Shrivastava et al.,
2013). However, Kleinman et al. showed that the observed evolution of aerosol size distribution
was not consistent with a volume growth mechanism in which the semivolatile organic vapors are



expected to readily diffuse within the accumulation mode substrate. This could indicate that the
accumulation mode particles over Mexico City were highly viscous and did not reach equilibrium
with large gas-phase organic molecules during the observation period. This observation is
consistent with our experimental results that toluene-derived SOM is highly viscous at RH < 20%
and Fig. 6, which shows that the median RH in Mexico City often falls below 20% in March.
However, it should be noted that the particulate matter over Mexico City is likely more chemically
complex than the SOM used in this study.

## 6 Conclusions

We investigated the RH-dependent viscosities at room temperature of SOM particles produced
from toluene photo-oxidation with the mass concentration of $60 - 1000$ μg·m$^{-3}$. A bead-mobility
technique showed the viscosities of the toluene-derived SOM increased from ~0.2 to ~129 Pa·s as
RH decrease from 89.9 to 60.7%. This indicates that the toluene-derived SOM particles are a liquid
at RH > 60%. The RH range for liquid-to-semisolid is in good agreement with Bateman et al.
(2015) who showed the liquid-to-semisolid phase transition of these particles in the range of 60-
80% RH. A poke-flow technique combined with fluid simulations showed the viscosities increased
from approximately $7.8 \times 10^3$ to $6.3 \times 10^6$ Pa·s as RH decreases from 47.3 to 30.5%. For RH ≤
17%, the viscosities of the SOM were greater than or equal to $\sim 2 \times 10^8$ Pa·s, similar to or greater
than the viscosity of tar pitch. This suggests that the toluene-derived SOM particles are a semisolid
at 20 < RH ≤ 60%, and a semisolid or a solid at RH ≤ 17%. Using the viscosity data and the Stokes-
Einstein equation, the diffusion coefficients of large gas-phase organic molecules within the
toluene-derived SOM particles were calculated to be $\sim 3 \times 10^{-8}$ to $\sim 2 \times 10^{-15}$ cm$^2$ s$^{-1}$ for RHs from
89.9 to 30.5%, and is lower than $\sim 3 \times 10^{-17}$ cm$^2$ s$^{-1}$ for RH ≤ 17%. Mixing time by diffusion of
large organic molecules within 200 nm toluene-derived SOM particles was calculated to be less
than 0.1 h at RH < 47.3%, 0.5 - 5 h at 30.5% RH, and greater than ~ 100 h at RH ≤ 17%.
To apply the results of the viscosities, diffusion coefficients, and mixing time of the toluene-
derived SOM, we selected the top 15 most populous megacities. Based on the RH in the cities, and
if the organic particulate matter in megacities is similar to the toluene-derived SOM in this study,
in cities such as Kolkata, Istanbul, Dhaka, Tokyo, Shanghai, and Mumbai, mixing times during



extended periods of the year will be very short and homogeneously well-mixed particles can be
assumed. On the other hand, for certain times of the year in Delhi, Beijing, Mexico City, Cairo,
and Karachi, mixing times of large organic molecules in organic particulate matter may be long ($\geq$
100 hr), and the particles may not be well mixed in the afternoon (3:00 – 5:00 local time) during
certain times of the year.  These results are summarized in Fig. 7.
**Acknowledgments**
This work was supported by the Natural Sciences and Engineering Research Council of Canada.
Support from the USA National Science Foundation, the Atmospheric Science Research (ASR)
Program of the USA Department of Energy, and research funds for newly appointed professors of
Chonbuk National University in 2015 is also acknowledged.

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





1 Table 1. Experimental conditions for production and collection of toluene-derived SOM particles

2 using the oxidation flow reactor. Particles were collected onto hydrophobic substrates using an

3 electrostatic precipitator or a single stage impactor.

| SOM sample name | Toluene conc. (ppm) | Ozone conc. (ppm) | SOM mass conc. during production ($\mu g\ m^{-3}$) | OFR flow rate ($L\ m^{-1}$) | Collection time (hr) | Collection method |
|---|---|---|---|---|---|---|
| For bead-mobility experiments | | | | | | |
| Toluene 1 | $1.0 \pm 0.1$ | $30 \pm 3$ | 600-1000 | $7.0 \pm 0.5$ | 48 | Electrostatic precipitator |
| Toluene 2 | $1.0 \pm 0.1$ | $30 \pm 3$ | 600-1000 | $7.0 \pm 0.5$ | 48 | Electrostatic precipitator |
| Toluene 3 | $0.1 \pm 0.01$ | $30 \pm 3$ | 60-100 | $7.0 \pm 0.5$ | 12 | Impactor |
| Toluene 4 | $0.1 \pm 0.01$ | $30 \pm 3$ | 60-100 | $7.0 \pm 0.5$ | 19 | Impactor |
| For poke-flow experiments | | | | | | |
| Toluene 5 | $1.0 \pm 0.1$ | $30 \pm 3$ | 600-1000 | $7.0 \pm 0.5$ | 96 | Electrostatic precipitator |
| Toluene 6 | $1.0 \pm 0.1$ | $30 \pm 3$ | 600-1000 | $7.0 \pm 0.5$ | 96 | Electrostatic precipitator |
| Toluene 7 | $0.1 \pm 0.01$ | $30 \pm 3$ | 60-100 | $7.0 \pm 0.5$ | 12.5 | Impactor |
| Toluene 8 | $0.1 \pm 0.01$ | $30 \pm 3$ | 60-100 | $7.0 \pm 0.5$ | 16 | Impactor |





Table 2. Physical parameters used to simulate material flow in the poke-flow experiments. $R$ and
$r$ indicate the radius of a tube and the radius of an inner hole, respectively.

|  | Slip length (nm) | Surface tension (mN m$^{-1}$) | Density (g cm$^{-3}$) | Contact angle (°) |
|---|---|---|---|---|
| Values used to calculate lower limit | 5[a] | 28[b] | 1.4[c] | 80 (if $r < 2R$), 100 (if $r > 2R$) |
| Values used to calculate upper limit | 10000[a] | 75[d] | 1.4[c] | 80 (if $r < 2R$), 100 (if $r > 2R$) |

[a] The range of slip length, which is the interactions between fluids and solid surfaces, is based on
literature data (Schnell, 1956; Churaev et al., 1984; Watanabe et al., 1999; Baudry et al., 2001;
Craig et al., 2001; Tretheway and Meinhart, 2002; Cheng and Giordano, 2002; Jin et al., 2004;
Joseph and Tabeling, 2005; Neto et al., 2005; Choi and Kim et al., 2006; Joly et al., 2006; Zhu et
al., 2012; Li et al., 2014).
[b] The lower limits of the surface tension of toluene-derived SOM were determined as 28 mN m$^{-1}$,
the surface tension of liquid toluene at 293 K (Adamson and Gast, 1997).
[c] Ng et al., 2007
[d] The upper limits of the surface tension of toluene-derived SOM were determined as 75 mN m$^{-1}$,
the surface tension of pure water at 293 K (Engelhart et al., 2008).
[e] Contact angle of the toluene-derived SOM on a substrate measured by 3-D fluorescence confocal
microscope ranged from 80 - 100°. The relationship of viscosity and contact angle depends on the
ratio of the radius of a tube, $R$, to the radius of an inner hole, $r$ (Grayson et al., 2015a).



1  Table 3. Months when the afternoon RH in the 15 most populous megacities either does not go

2  below 45 % (at the 10th percentile level) or is 17% or lower (at the 10th percentile level). "None"

3  indicates the RH does not meet either of these criteria.

| Megacity | Months when the afternoon RH (at the 10th percentile level) does not go below 45 % | Months when the afternoon RH (at the 10th percentile level) is 17 % or lower |
|---|---|---|
| Tokyo | Jun. – Sep. | none |
| Delhi | none | Mar. – Jun. |
| Shanghai | Jun. – Sep. | none |
| Mexico City | none | Jan. – May, Dec. |
| Sao Paulo | Jan. | none |
| Mumbai | May – Sep. | none |
| Osaka | Jul. | none |
| Beijing | none | Jan. – May, Nov. – Dec. |
| New York | none | none |
| Cairo | none | Mar. – May |
| Dhaka | Jun. – Oct. | none |
| Karachi | Jun. – Aug. | Jan. – Apr., Oct. – Dec. |
| Buenos Aires | none | none |
| Kolkata | May – Oct. | none |
| Istanbul | Jan. – Feb., Oct. – Dec. | none |



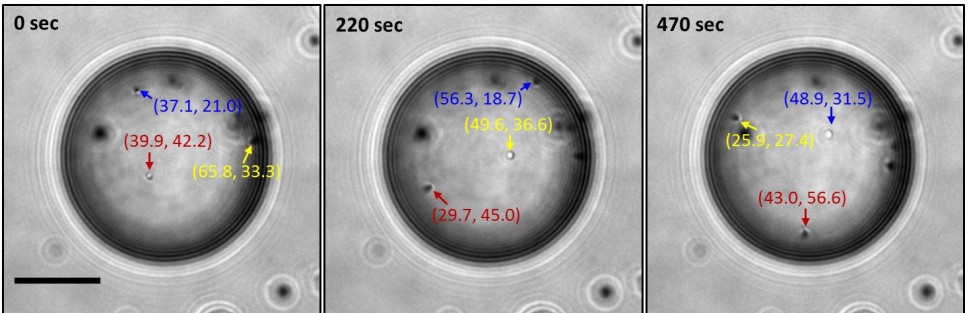

3    **Fig. 1.** Optical images from typical bead-mobility experiments for a toluene-derived SOM particle

4    (Toluene sample 8 in Table 1) at 80% RH. Three different beads are labeled using colored arrows.

5    The $x$ and $y$ coordinates of these beads are also indicated.  Scale bar: 20 μm.





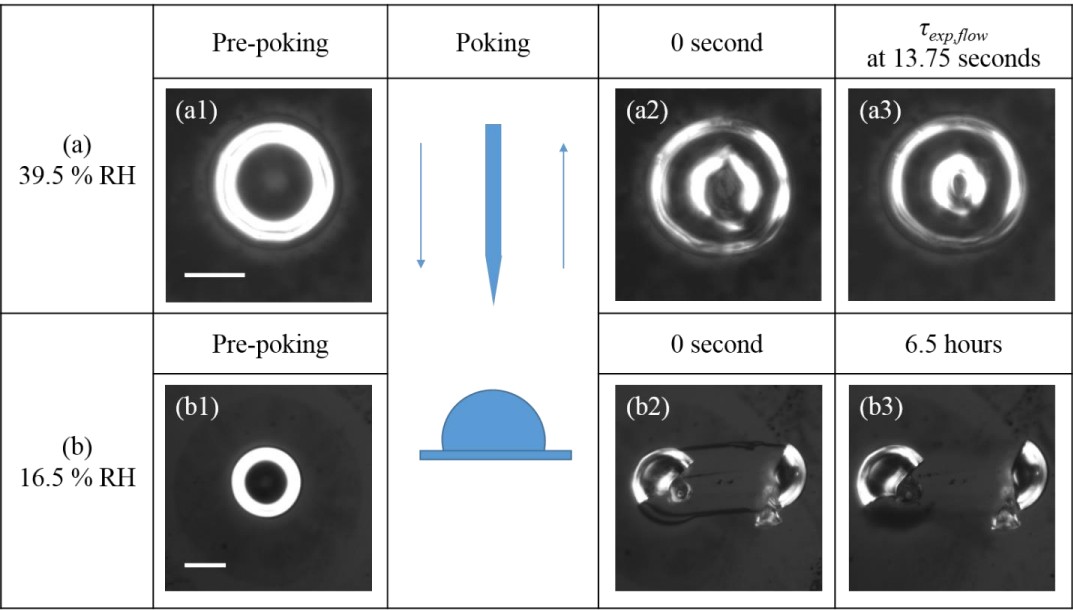

**Fig. 2.** Optical images of pre-poking, poking, and post-poking from typical poke-flow experiments
on toluene-derived SOM particles (toluene sample 6) at (a) 39.5% RH and (b) 16.5% RH. Panel
a1 and b1; pre-poking, Panel a2 and b2: post-poking immediately after the needle has been
removed (time set = 0 s), Panel a3; the experimental flow time, $\tau_{exp,\,flow}$, where the diameter of hole
has decreased to 50% of its initial size, and Panel b3; particles shatter and do not flow over a period
of 6.5 h. Size bar: 20 µm.





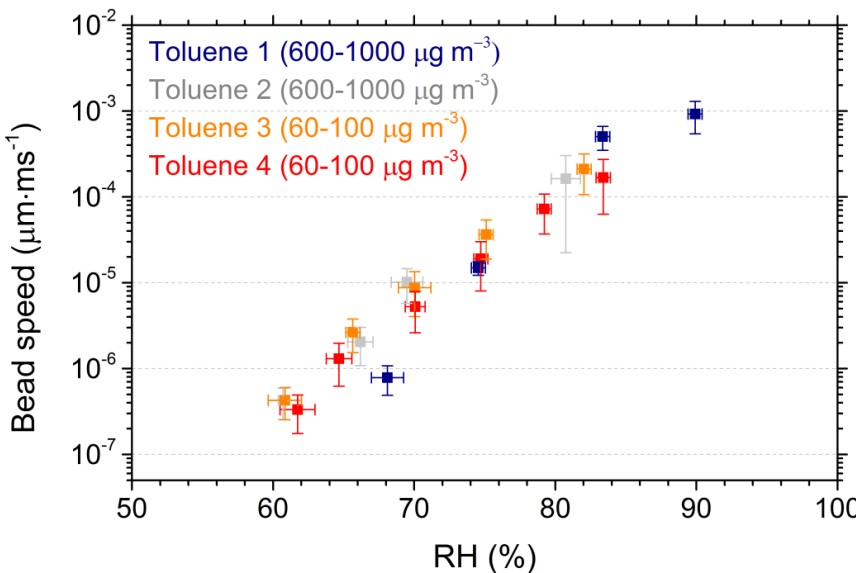

**Fig. 3.** Measured average bead speeds as a function of RH for different SOM samples (Toluene 1,
2, 3, and 4, see Table 1). The bead speeds of 3 - 10 beads were used to determine a mean bead
speed. The *x*-error bars represent the uncertainty in the RH measurements and the range of RH
values in a given experiment. The *y*-error bars represent the standard deviation of the measured
bead speeds.





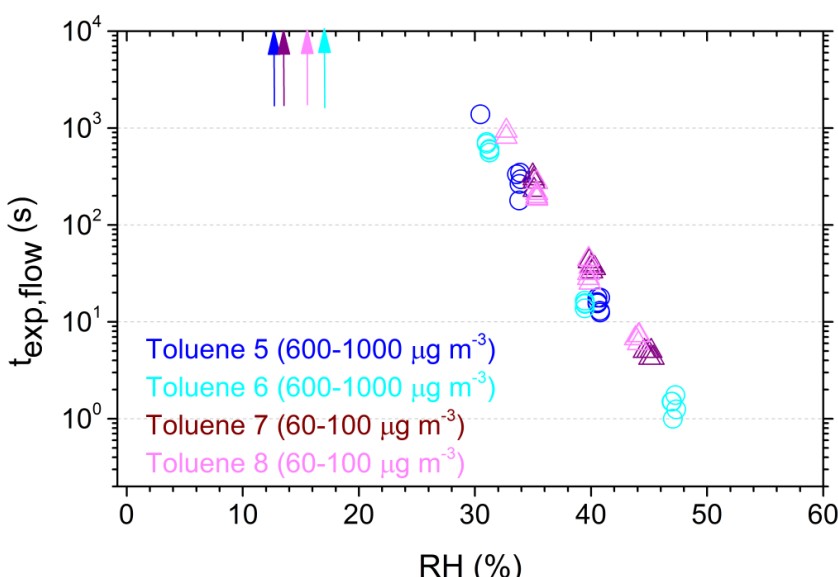

3   **Fig. 4.** Results from poke-flow experiments. $\tau_{exp,\,flow}$, where the diameter of hole has decreased to

4   50% of its initial size, measured for the different samples (Toluene 5, 6, 7, and 8, see Table 1).

5   The arrows indicate particles shattered at the given RH.





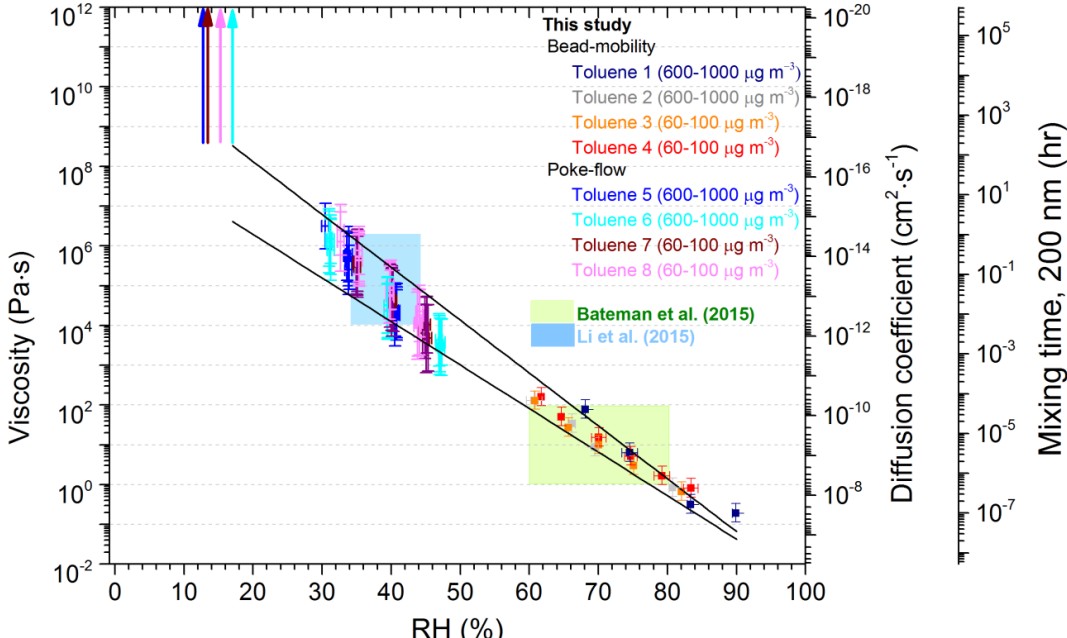

**Fig. 5.** Viscosities of toluene-derived SOM particles as a function of RH. For RH > 60% the viscosities were determined from the mean bead speeds (see Fig. 3) and a calibration line (Song et al., 2015). The $y$-error bars for RH > 60% represent the 95% prediction intervals from the calibration line. For RH < 60% the viscosities were calculated from the $\tau_{exp,\,flow}$ where $y$-error bars represent the calculated lower and upper limits of viscosity using the simulations. The $x$-error bars over the entire range of RH represent the range of RH values in a given experiments and the uncertainty in the RH measurements. The right $y$-axes present calculated diffusion coefficients of organic molecules in SOM using the Stoke-Einstein relation, and calculated mixing times within 200 nm particles due to bulk diffusion using Eq. (1). The black lines represent linear fits for the RH vs. log(lower viscosities) ($R^2$ = 0.958) and log(upper viscosities) ($R^2$ = 0.984) from the entire data set excluding the RH where particles cracked. Viscosity of toluene-derived SOM particles from Bateman et al. (2015) (green box) and Li et al. (2015) (blue box) is also included.





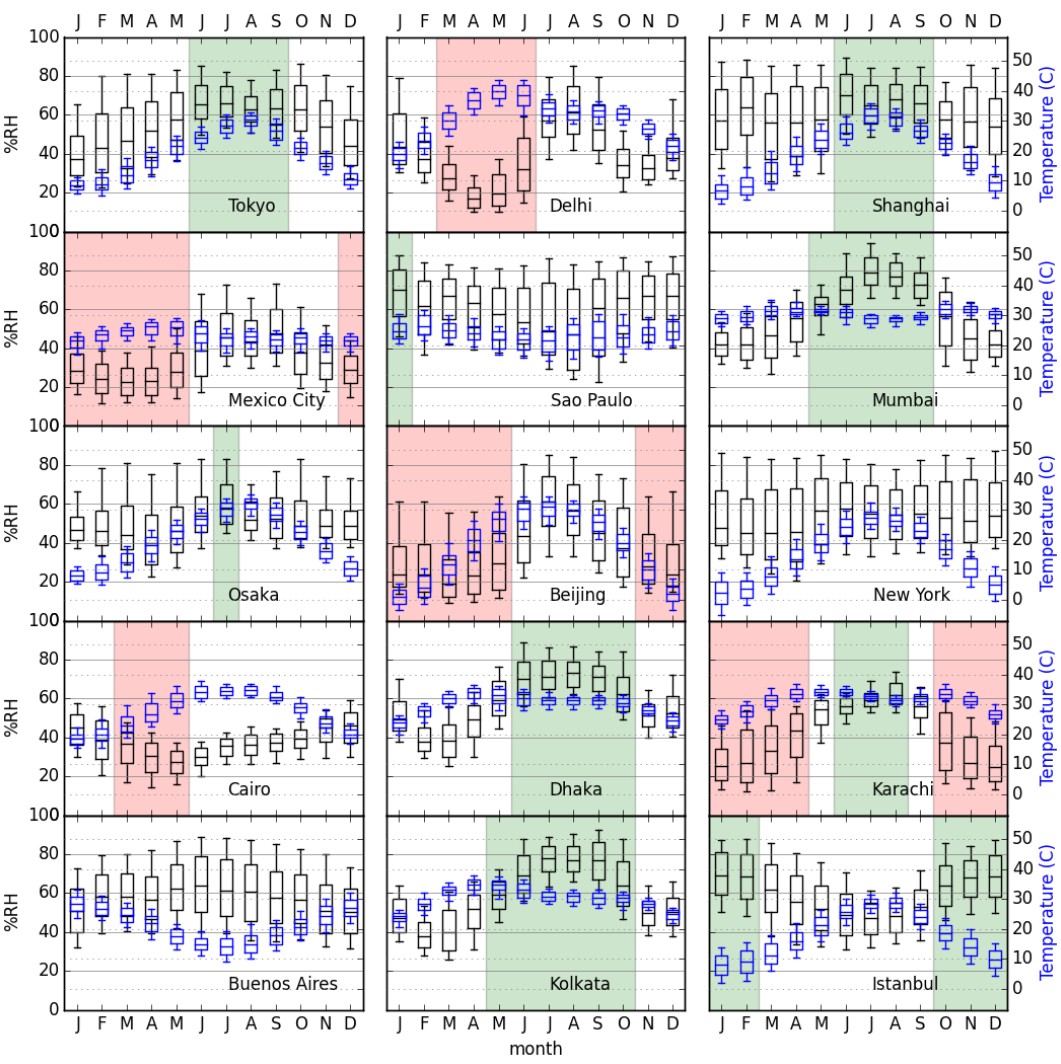

**Fig. 6.** Monthly average RH and temperature for the megacities of Tokyo, Delhi, Shanghai,
Mexico City, São Paulo, Mumbai, Osaka, Beijing, New York, Cairo, Dhaka, Karachi, Buenos
Aires, Kolkata, and Istanbul. For the stations, afternoon averages RH values (3-5 pm local time)
were retrieved from NOAA's National Climate Data Center for the years from 2004 to 2014. Boxes
show the median, 25th and 75th percentiles of 3-hr averages and the whiskers show the 10th and
90th percentiles. Green shading indicates that the afternoon RH (at the 10th percentile level) does





not go below 45% RH. Red shading indicates that the afternoon RH (at the 10th percentile level)
is 17% or lower.



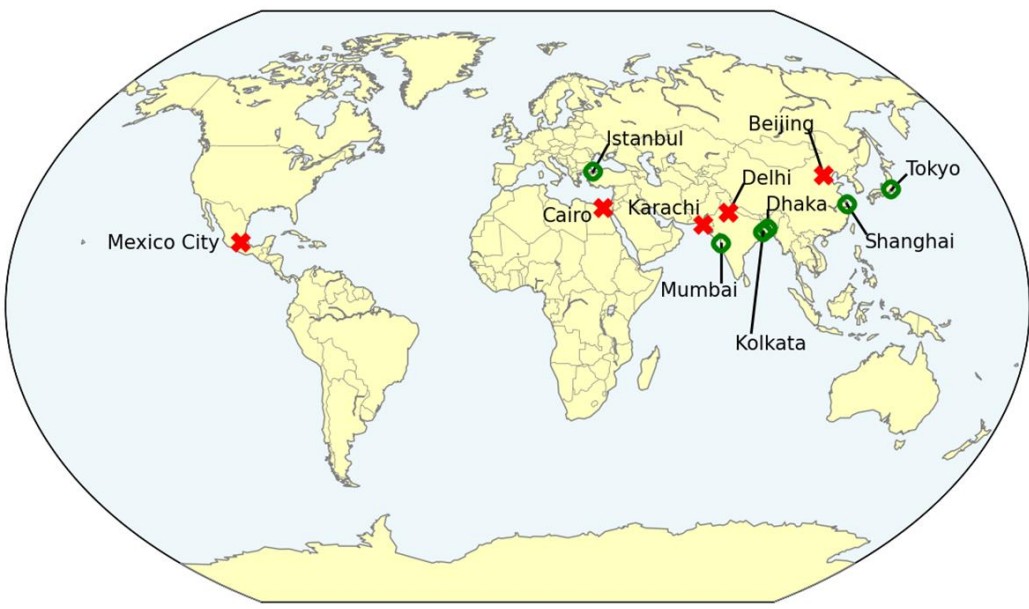

**Fig. 7**. Summary of conclusions reached after applying the results of the viscosities, diffusion
coefficients, and mixing time of the toluene-derived SOM to the top 15 most populous megacities.
Green circles indicates megacities where the afternoon RH at $10^{th}$ percentile does not go below
45 % RH for certain times of the year. In these cases, well-mixed particles can be assumed. Red
crosses indicates megacities where the afternoon RH at $10^{th}$ percentile is 17 % or lower for certain
times of the year. In these cases, the particles may not be well-mixed in the afternoon for certain
times of the year.