# Peer review of "Relative humidity-dependent viscosity of secondary organic"

_Atmospheric Chemistry and Physics, 2016_

## Referee Comment (RC1) · Anonymous Referee #1 · 25 Feb 2016

The manuscript describes measurements of the viscosity of secondary organic aerosol (SOA) samples derived from the oxidation of the anthropogenic precursor toluene using poke-flow and bead mobility techniques. The results are compared with previous measurements for the system derived from impactor measurements and uptake measurements of ammonia, and contrasted with previous measurements for SOA derived from typical biogenic precursors. The manuscript is concisely and clearly written, and the arguments presented are largely substantiated by the data reported. I have only a few minor comments suggesting possible revisions before the manuscript can be accepted for publication.

[Figure]

- My most significant comment is on the interpretation of the viscosities at low RH (below 20 %) and the conclusions the authors draw from these measurements. On page 12 of the manuscript, the authors state "Hence, the number of cases classified as having particles not well-mixed based on the viscosity data presented here and the Stokes-Einstein relationship should be considered as an upper limit." Earlier in the manuscript the authors highlight that the viscosity measurements made at such low RHs are only lower limits, with fluid flow and the recovery in particle shape in the poke-flow measurements requiring longer time than could be reasonably measured. Therefore, categorically stating that the diffusional mixing times at these low RHs provide an "upper limit" of the equilibration timescale seems to neglect that the viscosities could be orders of magnitude higher than the authors are assuming. Indeed, the authors state in the abstract that the diffusion constant is LOWER than $\sim 3 \times 10^{-17}$ cm2 s-1 for RH < 17%. So, although I would agree that the trend in viscosities from measurements at higher RHs would suggest that their conclusions about the limitations of slow mass transport being irrelevant for most geographical locations are probably reasonable, the data do not actually categorically prove this. Because of this I think the authors should add this caveat to their interpretation of the atmospheric relevance, both in the abstract and in the discussion/conclusions.

- Equilibration time before measurements are made, top of page 7: The authors state "These times are sufficient for the particles to equilibrate with the surrounding water vapor based on recent measurements of diffusion coefficients of water within SOM (Price et al., 2015)." The measurements the authors refer to are for alpha-pinene SOM. Thus, the timescale they must leave toluene SOA to equilibrate does not necessarily follow from this reference. A comment should be included by the authors on this.

- Uncertainties, top of page 9: The authors state "The uncertainty in the viscosity of approximately two orders of magnitude arises from the uncertainties in the physical parameters used in the simulations." The authors should say more explicitly what these uncertainties are.

- The authors present their new measurements of inferred diffusion constants from Stokes Einstein as being consistent with previous measurements, for example those of Li. Li et al. report diffusion constants of $10^{-17.5}$ m2 s-1 at 35-45 % RH. Given Stokes Einstein should fail to a similar degree for ammonia and water, it would suggest to me that the diffusion constants for semi-volatile organic molecules would be considerably lower than this at 35-45 % RH and even lower at RHs below 20 %. This is consistent I believe with the authors conclusions. However, they frequently switch between quoting diffusion constants in units of m2 s-1 and cm2 s-1, making it very confusing for the reader and difficult to verify the claims without very careful consideration. The authors should use one unit throughout.

- On page 10, the authors state: "However, the Stokes-Einstein relationship should give reasonable estimations of diffusion rates for large organic molecules for conditions not close to the glass transition temperature of the matrix". This is further supported by a recent study and review of recent data provided by Reid and coworkers on the effective volatility of a semi-volatile organic component from organic aerosol of varying viscosity (Chem. Sci., 2016, 7, 1298-1308, DOI: 10.1039/C5SC03223G). Referring to this paper could support further the argument.

---

## Referee Comment (RC2) · Anonymous Referee #2 · 24 Mar 2016

Review of Song et al., ACPD, 2016

The authors have measured relative humidity-dependent viscosity of toluene SOA after a series of publications on viscosity measurements of a-pinene and isoprene SOA. This group is currently one of the only groups that can achieve such challenging measurements by combining two different unique experimental methods of a bead-mobility technique and a poke-flow technique. Based on viscosity measurements and RH observations in major urban cities in the world, they estimated whether anthropogenic SOA particles in these cities would be well-mixed or not. The experiments seem to

be conducted well and the manuscript is written clearly, but some analysis/discussions should be deepened/expanded as below. I have several major comments that should be addressed and implemented in the revised manuscript before publication in ACP.

Major comments:

- Toluene is assumed to be a good proxy of anthropogenic precursors (P10, L28, without references). While this assumption might be fine, more justifications would be necessary: emission and ambient concentrations of toluene and other anthropogenic precursors are missing and not discussed sufficiently. Thus, general conclusion of phase state of OA in megacities (last sentence of abstract) sounds not convincing to me. The authors could look into emission inventories to check whether toluene is indeed a major precursor in selected major cities. Alkane, alkene and other aromatic compounds may be also important anthropogenic precursors and SOA generated by these precursors may have different viscosities. For example, a recent study showed that dodecane SOA may be less viscous and naphthalene SOA may be more viscous, compared to pinene and isoprene SOA (Berkemeier et al., ACP, 14, 12513, 2014). Please expand discussions on this issue.

- Moreover, as the authors also mentioned in P12, L15-20, I think that the temperature effect on viscosity is so important and probably as large as the RH effect (e.g., Koop et al., PCCP, 2011) that it is not easy to draw general conclusion of anthropogenic SOA phase state by measurements of toluene SOA only at room temperature. As shown in Fig. 6, most cities (except tropics) have lower temperature than 20 C in winter.

- As pointed out in the text, organic aerosols are likely to be internally mixed with inorganics such as sulfate and nitrate (P11, L2). This and also other groups have shown by experimental and modeling studies that a liquid-liquid phase separation is expected if an O:C ratio of an organic-rich phase is low, whereas a mixed one phase may be likely for high O:C. Have you measured an O:C ratio of toluene SOA? Fig. 7 in Lambe et al. AMT, 4, 445, 2011 showed that O:C ratio of OFR-generated toluene SOA

can be as high as 1.0. This issue should be discussed.

- What was RH in an OFR chamber during SOA formation? Was it all at the same RH? A recent study has shown that as the RH at which the a-pinene SOA is formed increases, there is a decrease in viscosity, and SOA that is formed dry and subsequently humidified remains solid to high RH (Kidd et al., PNAS, 111, 7552, 2014). This potential RH effect needs to be discussed.

- Water acts as plasticizer and plays a key role in determination of phase state. It would be interesting to plot viscosity against water mass fraction in SOA, which can be estimated using a hygroscopicity parameter kappa (e.g., Lambe et al., AMT, 4, 445, 2011; Hildebrandt Ruiz et al., ACP, 15, 8301, 2015).

Minor comments:

- P3, L11: The model used in Riipinen et al., 2011 did not treat bulk diffusion and they have rather emphasized an importance of gas diffusion and condensation. Thus, this reference seems not to be appropriate here. Instead, the authors could cite Shiraiwa et al., PNAS, 110, 11746, 2013.

- P4, L16: Loza et al., EST, 47, 6173, 2013 conducted also very similar experiments that may be worth included.

- P10, L7, L10: "However" is used twice in a row. Please refine.

---

## Author Comment (AC1) · 7 May 2016

The manuscript describes measurements of the viscosity of secondary organic aerosol (SOA) samples derived from the oxidation of the anthropogenic precursor toluene using poke-flow and bead mobility techniques. The results are compared with previous measurements for the system derived from impactor measurements and uptake measurements of ammonia, and contrasted with previous measurements for SOA derived from typical biogenic precursors. The manuscript is concisely and clearly written, and the arguments presented are largely substantiated by the data reported. I have only

a few minor comments suggesting possible revisions before the manuscript can be accepted for publication.

[1] My most significant comment is on the interpretation of the viscosities at low RH (below 20 %) and the conclusions the authors draw from these measurements. On page 12 of the manuscript, the authors state "Hence, the number of cases classified as having particles not well-mixed based on the viscosity data presented here and the Stokes-Einstein relationship should be considered as an upper limit." Earlier in the manuscript the authors highlight that the viscosity measurements made at such low RHs are only lower limits, with fluid flow and the recovery in particle shape in the poke flow measurements requiring longer time than could be reasonably measured. Therefore, categorically stating that the diffusional mixing times at these low RHs provide an "upper limit" of the equilibration timescale seems to neglect that the viscosities could be orders of magnitude higher than the authors are assuming. Indeed, the authors state in the abstract that the diffusion constant is LOWER than $3 \times 10$ЁЕ-17 cm2 s-1 for RH < 17%. So, although I would agree that the trend in viscosities from measurements at higher RHs would suggest that their conclusions about the limitations of slow mass transport being irrelevant for most geographical locations are probably reasonable, the data do not actually categorically prove this. Because of this I think the authors should add this caveat to their interpretation of the atmospheric relevance, both in the abstract and in the discussion/conclusions.

[A1] To address the referee's comment, we have done the following.

1. Added the following statement to the abstract, "As a starting point for understanding the mixing times of large organic molecules in organic particulate matter over cities, we applied the mixing times determined for toluene-derived SOM particles to the world's top 15 most populous megacities."

2. Added the following to Section 5: "Due to these caveats, the analysis below should be consider as a starting point for understanding the mixing times of large organic

molecules in organic particulate matter over megacities. Additional studies are needed to explore the implications of the caveats discussed above."

3. Removed the statement "Hence, the number of cases classified as having particles not well-mixed based on the viscosity data presented here and the Stokes-Einstein relationship should be considered as an upper limit".

[2] Equilibration time before measurements are made, top of page 7: The authors state "These times are sufficient for the particles to equilibrate with the surrounding water vapor based on recent measurements of diffusion coefficients of water within SOM (Price et al., 2015)." The measurements the authors refer to are for alpha-pinene SOM. Thus, the timescale they must leave toluene SOA to equilibrate does not necessarily follow from this reference. A comment should be included by the authors on this.

[A2] This is a good point. To address the referee's comment, we have now added the following text in Section 2.3. "These times should be sufficient for the particles to equilibrate with the surrounding water vapor based on recent measurements of diffusion coefficients of water within the water-soluble component of -pinene-derived SOM (Price et al., 2015). For example, the time to equilibrate with the surrounding of water vapor was calculated to be 25.3 min at 10 % RH based on diffusion coefficients of water within the water-soluble component of -pinene-derived SOM (Price et al., 2015). These diffusion coefficients should be applicable to SOM derived from toluene studied here, since both SOM have similar viscosities as a function of RH (compare Fig. 2 in Renbaum-Wolff et al. (2013a) with Fig. 5 below)."

[3] Uncertainties, top of page 9: The authors state "The uncertainty in the viscosity of approximately two orders of magnitude arises from the uncertainties in the physical parameters used in the simulations." The authors should say more explicitly what these uncertainties are.

[A3] To address the referee's comments we have modified this sentence to the following: "The uncertainty in the viscosity of approximately two orders of magnitude

arises from the uncertainties in the physical parameters used in the simulations (i.e. slip length, surface tension, density and contact angle). Of these parameters, the slip length contributed the most to the uncertainty in the viscosity."

[4] The authors present their new measurements of inferred diffusion constants from Stokes Einstein as being consistent with previous measurements, for example those of Li. Li et al. report diffusion constants of 10ËĘ-17.5 m2 s-1 at 35-45 % RH. Given Stokes Einstein should fail to a similar degree for ammonia and water, it would suggest to me that the diffusion constants for semi-volatile organic molecules would be considerably lower than this at 35-45 % RH and even lower at RHs below 20 %. This is consistent I believe with the authors conclusions. However, they frequently switch between quoting diffusion constants in units of m2 s-1 and cm2 s-1, making it very confusing for the reader and difficult to verify the claims without very careful consideration. The authors should use one unit throughout.

[A4] We now use the same unit (cm2 s-1) through the manuscript. Also it should be kept in mind that in Li et al. it was assumed that the overall rate of reaction was limited by the rate at which carboxylic acids diffused to the surface region of the particle, not the rate of diffusion of ammonia in the particles. The relevant section of the current manuscript has been modified to make this clear. The modified text is included below: "Li et al. (2015) previously estimated the diffusion coefficient of carboxylic acids within toluene-derived SOM from measurements of reactive uptake of NH3. They estimated a diffusion coefficient for carboxylic acids of 10-13.5±0.5 cm2Âůs‑1 for RHs between 35 and 45% using SOM mass concentrations of 44 to 125 $\mu$g m‑3. If a hydrodynamic radius of 0.1 - 1.5 nm is assumed for the carboxylic acids (Li et al., 2015), viscosity of 1 104 − 2 106 PaÂůs is calculated using the Stokes-Einstein equation (blue box in Fig. 5), consistent with our measurements."

[5] On page 10, the authors state: "However, the Stokes-Einstein relationship should give reasonable estimations of diffusion rates for large organic molecules for conditions not close to the glass transition temperature of the matrix". This is further supported

by a recent study and review of recent data provided by Reid and coworkers on the effective volatility of a semi-volatile organic component from organic aerosol of varying viscosity (Chem. Sci., 2016, 7, 1298-1308, DOI: 10.1039/C5SC03223G). Referring to this paper could support further the argument.

[A5] Thank you for the reference. It has been added to the revised manuscript.

---

## Author Comment (AC2) · 7 May 2016

The authors have measured relative humidity-dependent viscosity of toluene SOA after a series of publications on viscosity measurements of a-pinene and isoprene SOA. This group is currently one of the only groups that can achieve such challenging measurements by combining two different unique experimental methods of a bead-mobility technique and a poke-flow technique. Based on viscosity measurements and RH observations in major urban cities in the world, they estimated whether anthropogenic SOA particles in these cities would be well-mixed or not. The experiments seem to

be conducted well and the manuscript is written clearly, but some analysis/discussions should be deepened/expanded as below. I have several major comments that should be addressed and implemented in the revised manuscript before publication in ACP.

Major comments:

[1] Toluene is assumed to be a good proxy of anthropogenic precursors (P10, L28, without references). While this assumption might be fine, more justifications would be necessary: emission and ambient concentrations of toluene and other anthropogenic precursors are missing and not discussed sufficiently. Thus, general conclusion of phase state of OA in megacities (last sentence of abstract) sounds not convincing to me. The authors could look into emission inventories to check whether toluene is indeed a major precursor in selected major cities. Alkane, alkene and other aromatic compounds may be also important anthropogenic precursors and SOA generated by these precursors may have different viscosities. For example, a recent study showed that dodecane SOA may be less viscous and naphthalene SOA may be more viscous, compared to pinene and isoprene SOA (Berkemeier et al., ACP, 14, 12513, 2014). Please expand discussions on this issue.

[A1] To address the referee's comment, we have modified the first paragraph in section 5. The modified paragraph is reproduced below: "Several caveats should be kept in mind when applying the mixing times discussed earlier to particles over megacities. First, organic particulate matter over megacities are most likely more complicated than toluene-derived SOM. Toluene and other aromatics can account for a large fraction of nonmethane hydrocarbon emission in urban environments (Singh et al., 1985; Na et al., 2005; Suthawaree et al., 2012) and toluene and aromatics are thought to be one of the main sources of SOM particles in urban environments (Odum et al., 1997; Schauer et al., 2002a; 2002b; Vutukuru et al., 2006; Velasco et al., 2007; de Gouw et al., 2008; Velasco et al., 2009; Gentner et al., 2012; Liu et al.. 2012; Hayes et al., 2015). Nevertheless, large alkanes and unspeciated nonmethane organic gases also likely play a role in SOM formation in urban environments. Second, the toluene-derived SOM was

generated using relatively large mass concentrations of particles (60 – 1000 $\mu$g m-3). The good agreement between our results and the results from Bateman et al. (2015) and Li et al. (2015), which were carried out with a mass concentration of 30 – 1000 $\mu$g m-3, suggests that for toluene-derived SOM the viscosity is not strongly dependent on the mass concentration of organics used to generated the SOM, but additional studies are needed to confirm this. Third, as mentioned above, the Stokes-Einstein equation was used to estimate diffusion coefficients and hence mixing times, and this equation can underestimate diffusion coefficients close to the glass transition temperature. Due to these caveats, the analysis below should be consider as a starting point for understanding the mixing times of large organic molecules in organic particulate matter over megacities. Additional studies are needed to explore the implications of the caveats discussed above."

[2] Moreover, as the authors also mentioned in P12, L15-20, I think that the temperature effect on viscosity is so important and probably as large as the RH effect (e.g., Koop et al., PCCP, 2011) that it is not easy to draw general conclusion of anthropogenic SOA phase state by measurements of toluene SOA only at room temperature. As shown in Fig. 6, most cities (except tropics) have lower temperature than 20 C in winter.

[A2] Here the referee is raising the issue of temperature effect on viscosity of aerosol particles. To address the referee's comments in the revised manuscript we have limited the analysis to months when the median afternoon temperature is within 5 K of the temperatures used in the viscosity measurements (i.e. 290 to 300 K). The following text has also been added to the manuscript: "In addition to relative humidity, viscosity can depend strongly on temperature (Champion et al., 1997; Koop et al., 2011). For example, the viscosity of solutions of sucrose and water may increase by two to three orders of magnitude for a 10 K decrease in temperature close to the glass transition temperature (Champion et al., 1997). However, the effect of temperature on the viscosity of toluene-derived SOM has not been quantified. As a result, we have limited the current analysis to months when the median afternoon temperature is within 5 K

of the temperatures used in the viscosity measurements (i.e. 290 to 300 K). The fact that the median afternoon temperature is often below 290 K, highlights the need for low-temperature viscosity measurements."

[3] As pointed out in the text, organic aerosols are likely to be internally mixed with inorganics such as sulfate and nitrate (P11, L2). This and also other groups have shown by experimental and modeling studies that a liquid-liquid phase separation is expected if an O:C ratio of an organic-rich phase is low, whereas a mixed one phase may be likely for high O:C. Have you measured an O:C ratio of toluene SOA? Fig. 7 in Lambe et al. AMT, 4, 445, 2011 showed that O:C ratio of OFR-generated toluene SOA can be as high as 1.0. This issue should be discussed.

[A3] To address the referee's comments the following text has been added to the revised manuscript: "Mass concentrations of SOM particles in the OFR were 60-100 $\mu$g m-3 and 600-1000 $\mu$g m-3 for the two different experimental conditions (see Table 1). For the mass concentration of 60-100 $\mu$g m-3, the oxygen-to-carbon (O:C) ratio was 1.08, calculated from the AMS mass spectra following the approach of Chen et al. (2011). This value can be compared with the O:C values ranged from 0.9 to 1.3 measured for toluene-derived SOM generated in a similar OFR (Lambe et al., 2015)."

[4] What was RH in an OFR chamber during SOA formation? Was it all at the same RH? A recent study has shown that as the RH at which the a-pinene SOA is formed increases, there is a decrease in viscosity, and SOA that is formed dry and subsequently humidified remains solid to high RH (Kidd et al., PNAS, 111, 7552, 2014). This potential RH effect needs to be discussed.

[A4] To address the referee's comment, the following text has been added to Section 2.1 of the revised manuscript.

"The relative humidity inside the reactor was held constant at 13 +- 3 %. A recent study has shown that the viscosity of -pinene-derived SOM is dependent on the RH at which the SOM is generated (Kidd et al., 2014). Additional studies are needed to explore this

potential RH effect on the viscosity of toluene-derived SOM."

[5] Water acts as plasticizer and plays a key role in determination of phase state. It would be interesting to plot viscosity against water mass fraction in SOA, which can be estimated using a hygroscopicity parameter kappa (e.g., Lambe et al., AMT, 4, 445, 2011; Hildebrandt Ruiz et al., ACP, 15, 8301, 2015).

[A5] Thank you for the suggestion. To address the referee's comment we have added Vwet/Vdry as a secondary x-axis to Fig. 5, where Vdry is the volume of SOM at 0% RH and Vwet is the volume of the SOM after taking up water at a given RH. In addition the text below has been added to the manuscript:

"The strong dependence of viscosity on RH shown in Fig. 5 can be understood by considering the hygroscopic nature of the SOM. To illustrate this point in Fig. 5 viscosity is also plotted versus Vwet/Vdry of the SOM (secondary x-axis), where Vdry is the volume of SOM at 0% RH and Vwet is the volume of the SOM after taking up water at a given RH. Vwet/Vdry was calculated with the following equation (Petters and Kreidenweis, 2008; Pajunoja et al., 2015):

$V\_wet/V\_dry = \kappa/(100/RH-1)+1$ (Eq. 1)

where is the hygroscopic parameter. A hygroscopic parameter of 0.15 was assumed, consistent with previous measurements for toluene-derived SOM (Ruiz et al., 2015). Equation (1) neglects the Kelvin effect, which is a reasonable assumption for the large particles used in our studies. Fig. 5 illustrates that the water content (top x-axis) of the particles plays a key role in regulating the viscosity. ."

Minor comments:

[6] P3, L11: The model used in Riipinen et al., 2011 did not treat bulk diffusion and they have rather emphasized an importance of gas diffusion and condensation. Thus, this reference seems not to be appropriate here. Instead, the authors could cite Shiraiwa et al., PNAS, 110, 11746, 2013.

[A6] We have now corrected the reference.

[7] P4, L16: Loza et al., EST, 47, 6173, 2013 conducted also very similar experiments that may be worth included.

[A7] The paper has been added.

[8] P10, L7, L10: "However" is used twice in a row. Please refine.

[A8] It has been revised.